# Economics of vaccination against diarrhoea and respiratory diseases in French cow calves' systems: A modelling approach

Ahmed Ferchiou[1]*, Anthony Giacomini[1], Nicolas Herman[2], Guillaume Lhermie[1,3], Didier Raboisson[1]

1 UMR ASTRE, Université de Toulouse, CIRAD, INRAE, ENVT, Toulouse, France, 2 Clinique Vétérinaire Des Mazets, Les Mazets, Riom-ès-Montagnes, France, 3 Faculty of Veterinary Medicine, University of Calgary, Calgary, Canada

* ahmed.ferchiou@cirad.fr

## Abstract

To assess the economic benefits of vaccination against diarrhoea and respiratory diseases in cow-calf systems, a stochastic mechanistic mathematical programming bio-economic model was developed for a beef Charolais breed farm. The model reproduced herd dynamics on a weekly basis over a 10-year period, including animal growth, reproduction and diseases risks and control (treatments and vaccination). Three baseline risk levels of diarrhoea and respiratory disease were considered to simulate low, average, and high infection risks on the farm. The results demonstrated the model's ability to reproduce average farm dynamics, with diarrhoea prevalence ranging from 8% to 33% and respiratory diseases from 13% to 79%. On average, prevalences were decreased by 24% and 50%, respectively, due to vaccination, which also reduced the 2-day to 1-month mortality rates by 2–3% and 15–17%, respectively. The net value of vaccination was positive except in cases of low infection risk of diarrhoea and respiratory diseases: the return on investment (ROI) of vaccination was 0.4–0.5 (€1 invested yields €0.4–0.5) in the baseline scenario (middle infection risks), negative under low infection risks, and reached 1 (€1 invested yields €1) for high infection risks. The results also showed that farmers may earn €58–75 in additional revenue per extra hour of labour dedicated to vaccination in the medium-risk scenario. Despite the limited availability of data for cow-calf systems, this study demonstrated the economic benefit of vaccination in most combinations of scenarios and risks. The profitability of vaccination was challenged when low disease risks were considered, but in such cases, vaccination may serve as insurance against financial losses in case of health status deterioration.

**Data availability statement:** All relevant data are within the manuscript and its Supporting Information files.

**Funding:** The author(s) received no specific funding for this work.

**Competing interests:** The authors have declared that no competing interests exist.

## Introduction

In France, as recorded in the national identification database (BDNI), calf mortality before the age of one month reaches 6–10% in cow-calf farms and even 10–14% in dairy cattle. Apart from dystocia, the main causes of death during this period are neonatal diarrhoea and bovine respiratory diseases (BRD) [1]. Neonatal diarrhoea is a complex, multifactorial syndrome with a variety of infectious etiologies including bacteria (*Escherichia coli, Salmonella* spp.…), viruses (rotaviruses, coronaviruses…) and parasites (*Giardia duodenalis, Cryptosporidium muris…*) [2]. Similarly, BRD involves bacterial (*Mannheimia haemolytica, Pasteurella multocida…*) and viral agents (mainly Bovine Respiratory Syncytial Virus, Influenza…) as well as a complex set of physiological and environmental predisposing factors [3,4]. The prevention and management of neonatal diarrhoea and BRD involve reducing pathogen pressure and strengthening calves' natural defenses. Reducing pathogen pressure may include treatments, rigorous housing management in winter, herd management policies that prevent overcrowding, avoiding the mixing of calf batches of different ages, quarantining, disinfection, and continuous monitoring of herd health status. Strengthening calves' natural defenses involves ensuring their thermal and hygrometric comfort, minimizing stress, optimizing colostrum intake, and possibly implementing targeted vaccination [1].

Diarrhoea and BRD impose a significant economic burden on cow-calf systems. Previous studies have shown that losses associated with BRD account for more than 20% of French farmers' income losses, and the direct costs per animal for viral digestive disorders average €150–€175 per animal per year [5,6]. These costs arise from deaths, healthcare expenses, including antibiotics, and lost production [7,8].

Disease prevention is often reported as profitable, even though the evidence supporting this claim is limited [9,10]. Vaccination against neonatal diarrhoea and BRD may improve profitability and reduce antimicrobial use (AMU) while allowing better work planning. While neonatal diarrhoea and BRD are the primary health challenges, vaccination against at least one of these two conditions was reported in 36.5–52% of cases in surveys from 2016 and 2022 [11,12]. Farmers report several reasons for this low vaccination coverage, including cost, administration constraints, compliance with vaccination protocols, perceived efficacy, and poor information [13–15].

Therapeutic treatments include antibiotics, and their misuse is a driver contributing to the emergence of antibiotic resistance. A continued reduction in their use through improved non-specific or specific prevention measures is recommended [16,17]. To address this challenge in animal health, France implemented the Eco-antibio plan in 2011 [18]. The most recent phase of the plan (2023–2028) aims to strengthen disease prevention measures. In 2022, cattle were the primary consumers of antibiotics in France, with 111.16 tons of antibiotics consumed, accounting for 40% of national antibiotic sales. Since 2011, antibiotic exposure in cattle has decreased by 22.6%, but the overall level of antibiotic exposure has remained relatively stable since 2016 [19]. As in many other countries, the French cattle farming sector faces additional major concerns, including price volatility and the need to improve animal welfare.

Several bio-economic simulation models have been developed to support decision-making in animal health within beef cattle production systems, primarily addressing infectious diseases of parasitic, viral, or bacterial origin [20–22]. These models have been developed for various production systems, such as feedlot systems, cow-calf systems, or more extensive systems [23–26]. They are based on a compartmental representation of the population to model the infection dynamics of the disease under study.

To our knowledge, no holistic agent-based model representing all the functions of a cow-calf farming system and simulating a range of production diseases has been developed to represent beef cattle production systems. Such an approach would allow for the assessment of trade-offs between vaccination strategies for both respiratory and digestive disorders, while integrating a multi-criteria approach that considers economic outcomes, farmer workload, and antibiotic use.

The objective of this study was to assess the direct on-farm economic net value of vaccination against diarrhoea and respiratory diseases in cow-calf herds, while considering reductions in AMU and other benefits for farmers.

## Materials and methods

The integrated bio-economic modelling approach previously developed for dairy farming was adapted and tailored to beef cattle farming [27]. The resulting model was a biological simulation framework coupled with an economic optimization model, named BeefHealthSim. The biological model operated on a cow-week basis, incorporating weekly probabilities of events for all cows, including growth, reproduction, and diseases (Fig. 1).

The model provided a dynamic representation of the cow-calf herd. Briefly, from birth to death, each animal was characterized on a weekly basis by its physiological and production status (e.g., male calf, female calf, pregnant female, cow). This framework encompassed three core interconnected functions:

- Production (e.g., growth and reproduction),

- Diseases (which affect production – health damage),

- Treatments (as disease control – damage control).

### Growth function and related feeding principles

The growth of the animals was modeled using weekly live weight gain (ADG: Average Daily Gain), calculated separately for male and female calves to account for their differing birth weights and growth rates. The ADG was defined by age group (Table 1):

- For males: a growth phase from 0 to 210 days of age ($ADG_{<210M}$) and a finishing phase after 210 days until sale ($ADG_{>210M}$);

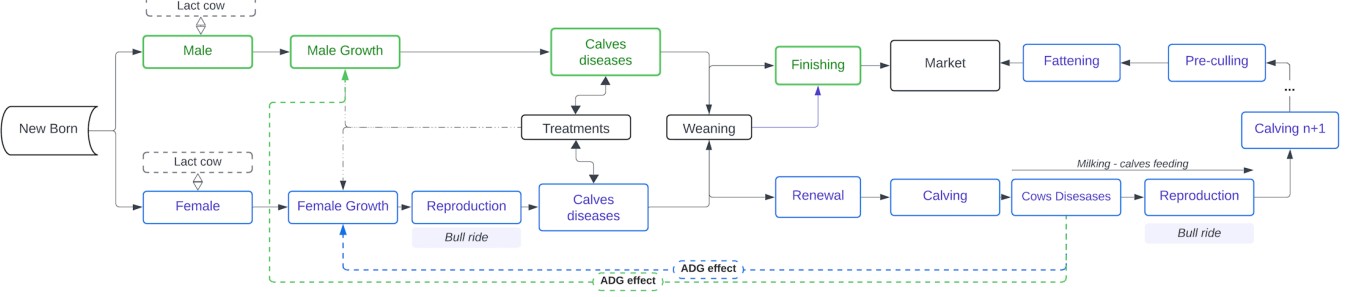

**Fig 1. Overview of the BeefHealthSimulator (BHS) biological model.**

**Table 1. Calibration for reproduction and weight gain.**

| Indicator (unit) | Male (SD) | Female (SD) | Ref. |
|---|---|---|---|
| Weight$_{Birth}$ (kg) | 48 (5) | 45 (5) | [9,28,29] |
| ADG$_{age<210d}$ (g/d) | 1150 (115) | 1000 (100) | |
| ADG$_{age>210d\_Males}$ (g/d) | 1350 (135) | .. | |
| ADG$_{age210d-280d\_Female}$ (g/d) | .. | 1130 (113) | |
| ADG$_{280-Pub}$ (g/d) | .. | 650 (65) | |
| ADG$_{Pub-Calv}$ (g/d) | .. | 540 (54) | |
| Weight$_{Obj}$ (kg) | 380 (19) | 300 (15) | |
| Weight$_{Pub}$ (kg) | .. | 380 (30) | [30] |
| Age$_{Pub}$ (wks) | .. | [52; 70] | [30] |
| Weight$_{Mat}$ (kg) | .. | [500; 520] | [30] |
| Age$_{Mat}$ (wks) | .. | [80; 108] | Authors |
| Anoest$_{Dur}$ (wks) | .. | 2 | [30] |
| P$_{MatingCycle1}$ | .. | 0.3 | Authors |
| P$_{MatingCycle>1}$ | .. | 0.65 | Authors |
| CoefBullAvail | .. | 0.8 | Authors |
| Preg$_{Dur}$ (wks) | .. | 40 (1) | BDNI* |
| AbortRate (%) | .. | 0.7% | [31] |
| ReplacementRate (%) | .. | 26% | [29] |
| CalvesMortRate (%) | 5% | 5% | BDNI* |
| Price sold calves (€/kg live-weight) | 3.9 | 3.5 | [28,32] |
| Price culled cows (€/kg carcass) | .. | 5.39 | |

\*: BDNI refers to the National Animal Identification Database, which ensures the registration, identification, and traceability of cattle movements. Ref. for References.

- For females: a growth phase from 0 to 210 days (ADG$_{<210F}$), an intermediate growth phase between 210 and 280 days (ADG$_{210-280F}$), a pre-puberty phase until puberty (ADG$_{280-PubF}$), and finally, a gestational phase until calving (ADG$_{Pub-Calv}$). After calving, the cow's body weight was simulated using the Van Arendonk equation (1985), which was calibrated with birth weight and mature weight parameters.

Male and female calves were managed similarly until weaning (Fig 1: Weaning). Males were weaned and sold on the same day, after reaching 300 days of age and achieving a target weight (Weight$_{ObjM}$). The decision to sell was made weekly, based on each animal's weight. Female calves were also weaned at 10 months of age, when they achieved their target weight (Weight$_{ObjF}$). The heaviest females, prioritized for herd renewal due to lower disease incidence (Fig 1: Renewal), were retained, while others were sold.

The feeding of calves was modeled by physiological stages, separately for males and females. Each stage was associated with a specific ADG and an average feeding cost (Table 2):

1) Milk-based feeding;

2) Introduction of hay and grass (lower feeding costs);

3) Introduction of concentrates;

4) Rearing without weaning (ad libitum concentrates for both males and females until sale).

## Reproduction function

The onset of puberty in heifers was modeled based on weight and age thresholds ($Age_{Pub}$; $Weight_{Pub}$). Similarly, the decision to start breeding heifers depended on weight and age ($Age_{Mat}$; $Weight_{Mat}$), with the goal of achieving a first calving at 36 months. The reproductive cycle was simulated with a weekly probability of occurrence after puberty ($P_{cycleCalv1}$). The probabilities of successful insemination and abortion were applied similarly to those for cows (Table 1 and Table 3).

For cows, after the postpartum anoestrus period ($Anoest_{Dur}$), a reproductive cycle was simulated based on a weekly probability of occurrence after calving ($P_{cycleCalv>1}$). At the end of the 3-week ovarian cycle, natural mating was modeled, with success rates varying depending on the cycle following anoestrus ($P_{MatingCycle1}$; $P_{MatingCycle>1}$). Bull availability was factored into the model as 1 bull per 25 cows; if there was more than 1 bull per 30 cows to be mated, mating success was adjusted using a coefficient (CoefBullAvail). For both cows and heifers, after successful mating, a gestation period ($Preg_{Dur}$) began, during which an abortion risk was simulated (AbortRate).

## Herd dynamics

Calvings were grouped into two seasons annually, occurring from August to October and from February to April. The number of heifers retained for herd renewal at weaning corresponded to the replacement rate (ReplacementRate) plus a 5% adjustment to account for mortality and culling of heifers before their first calving.

**Table 2. Daily cost for diet per category of age in months [9].**

| Age interval (months) | DailyFeedCost$_{Male}$ (€) | DailyFeedCost$_{Female}$ (€) |
|---|---|---|
| 0–4 | 0.1 | 0.1 |
| 4–5 | 0.4 | 0.4 |
| 5–6 | 0.84 | 0.84 |
| 6–weaning | 1.07 | 0.84 |
| weaning–24 | .. | 0.41 |
| 24–36 | .. | 0.51 |
| 36–calving | .. | 0.61 |
| cow | .. | 0.41 |

**Table 3. Probabilities of ovarian cycle (end of anoestrus) per week post-partum for primiparous ($P_{cycleCalv1}$) and multiparous ($P_{cycleCalv>1}$) cows [30].**

| Week post-partum | $P_{cycleCalv1}$ (SD) | $P_{cycleCalv>1}$ (SD) |
|---|---|---|
| 1–3 | 0 (0) | 0 (0) |
| 4 | 0 (0) | 0.05 (0.0025) |
| 5 | 0 (0) | 0.15 (0.0075) |
| 6 | 0.05 (0.0025) | 0.3 (0.015) |
| 7 | 0.15 (0.0075) | 0.5 (0.025) |
| 8 | 0.3 (0.015) | 0.7 (0.035) |
| 9 | 0.5 (0.025) | 0.9 (0.045) |
| 10 | 0.7 (0.035) | 1 (0.05) |
| 11 | 0.9 (0.045) | 1 (0.05) |
| 12 | 1 (0.05) | 1 (0.05) |

Heifers were culled for two primary reasons:

1) A weight reduction greater than 20% of the target (culling occurs during the breeding period or at any point thereafter);

2) Three unsuccessful mating attempts.

Cow culling occurred in two stages:

1) Pre-culling:

- A cow was pre-culled if it was not pregnant by the end of the breeding season;

- If it has experienced two consecutive abortions;

- If it aborted after 200 days of gestation, post-calving, or after the sale of the calf.

The number of pre-culls per year was evenly distributed across the two calving seasons in proportion to the number of calvings. Pre-culling did not exceed the number of heifers available for replacement (with herd stability prioritized). Pre-culled cows were fully culled after the calf was sold, weaned, or died following a 2-month fattening period.

2) Quota-based culling: if the annual culling quota was unmet, cows with more than six lactations were pre-culled first. If the quota remained unfulfilled, the oldest cows were culled.

**Diseases: Risks and impacts**

Diseases in suckler cows were considered during specific postpartum weeks. They included:

- Dystocia: with a base risk ($Risk_{Dys}$) and an additional risk if the calf was stillborn ($RR_{Dys\_IfStillbirth}$);

- Foetal membranes retention: base risk ($Risk_{FMR}$) with an additional risk if dystocia ($RR_{FMR\_IfDys}$) occurred;

- Metritis: base risk ($Risk_{Met}$) with additional risks if dystocia ($RR_{Met\_IfDys}$) or foetal membrane retention ($RR_{Met\_IfFMR}$) occurred;

- Endometritis: base risk ($Risk_{PVD}$) with an additional risk if metritis ($RR_{PVD\_IfMet}$) occurred.

These diseases lead to milk production losses, reducing average daily gain ($ADG_{IfDys}$, $ADG_{IfFMR}$ and $ADG_{IfMet}$), increased mortality risks ($RR_{Death\_IfDys}$ and $RR_{Death\_IfMet}$), and impairing reproductive performance ($RR_{CR}$: relative risk on conception rate).

Calf diseases were modeled weekly and included:

- Failure of passive immune transfer (FIT): base risk ($Risk_{FIT}$) with an additional risk if dystocia ($RR_{FIT\_IfDys}$) occurred;

- Diarrhoea: base risk ($Risk_{Dia}$) with an additional risk if FIT ($RR_{Dia\_IfFIT}$) occurred;

- Omphalitis: base risk ($Risk_{Omph}$) with additional risks for dystocia ($RR_{Omph\_IfDys}$) and male calves ($RR_{Omph\_IfMale}$);

- Septicaemia: base risk ($Risk_{Sept}$) with an additional risk if FIT ($RR_{Sept\_IfFIT}$) occurred;

- Respiratory diseases (BRD): base risk ($Risk_{BRD}$) with an additional risk if FIT ($RR_{BRD\_IfFIT}$) occurred.

These diseases were associated with reduced growth rates over varying periods ($ADG_{IfDia}$, $ADG_{IfOmph}$ and $ADG_{IfBRD}$) and additional mortality risks ($RR_{Death\_IfDia}$, $RR_{Death\_IfOmph}$, $RR_{Death\_IfSept}$ and $RR_{Death\_IfBRD}$). No effects on animal conformation due to disease were included (Table 4).

## Treatment functions

The disease control functions simulate treatments for cattle and their impact on health. Each treatment model was characterized by:

1) Composition: medications (e.g., antibiotics, anti-inflammatories) and intervention (e.g., consultation, surgery);

2) Expected effects: on disease recovery and risk of relapse;

3) Costs: additional labour, treatment costs, and veterinary fees.

For each disease, a probability of intervention ($P_{Trt}$) was simulated for either the farmer or the veterinarian. Treatment protocols were selected based on probabilities ($P_{Prot}$) (S1 Table in S1 File). Information on drug administration and composition was based on the Summary of Product Characteristics for authorized veterinary medicines (S2 Table in S1 File).

**Table 4. Risks of cows and calves' diseases and their consequences on production.**

| | Risk of the disease | | Consequence of the disease on the production | | | Ref. |
|---|---|---|---|---|---|---|
| Disease[1] | Risk[2] (week: value) | Relative Risks of diseases (RR) | ΔADG[3] (duration) | RR_Death (duration) | RR_CR[4] (duration) | |
| Dys | w1: 0.1 | $RR_{Dys\_IfStillbirth}=14.6$ | $\Delta ADG_{IfDys}=-10g$ (4 wks) | $RR_{Death\_IfDys}=3$ (1) | No | [9] |
| FMR | w1: 0.04 | $RR_{FMR\_IfDys}=3$ | $\Delta ADG_{IfFMR}=-130g$ (2 wks) | No | $RR_{CR\_IfFMR}=0.4$ (13 wks) | [9] |
| Met | w1: 0.02 w2: 0.01 | $RR_{Met\_IfDys}=1.4$ $RR_{Met\_IfFMR}=5$ | $\Delta ADG_{IfMet}=-100g$ (4 wks) | $RR_{Death\_IfMet}=1.4$ (4 wks) | $RR_{CR\_IfMet}=0.4$ (7 wks) | [9] |
| PVD | w3: 0.023 w4: 0.017 w5: 0.011 w6: 0.006 | $RR_{PVD\_IfMet}=2.3$ | No | No | $RR_{CR\_IfPVD}=0.69$ (13 wks) | [9] |
| FIT_Mod | w0: 0.10 | $RR_{FITMod\_IfDys}=1.5$ | .. | .. | .. | [33] |
| FIT_Sev | w0: 0.10 | $RR_{FITSev\_IfDys}=2.5$ | .. | .. | .. | [33] |
| Dia | w1: 0.04 w2: 0.09 w3: 0.07 w4: 0.03 | $RR_{Dia\_IfFIT}=1.5$ | $\Delta ADG_{IfDia}=-200g$ (6/2 wks)[5] | $RR_{Death\_IfDia}=5$ (Up to 1 wk age) 1.78 (2–5 wks age) 1.09 (age > 5 weeks) | .. | [33] |
| Omph | w1: 0.10 w2: 0.10 w3: 0.02 | $RR_{Omph\_IfDys}=2$ $RR_{Omph\_IfMale}=2$ | $\Delta ADG_{IfOmph}=-54g$ (1st case)/ -108g (>=2nd case) (24 wks)[5] | $RR_{Death\_IfOmph}=5$ (9 wks) | .. | [33] |
| Sept | w1: 0.02 w2: 0.005 w3: 0.002 w4: 0.001 | $RR_{Sept\_IfFITSev}=5$ | .. | $RR_{Death\_IfSept}=5$ (9 wks) | .. | [33] |
| BRD | w1 to w26: 0.02 per week | $RR_{BRD\_IfFITMod}=1.73$ $RR_{BRD\_IfFITSev}=2.27$ | $\Delta ADG_{IfBRD}=-40/-100/-200g$ (10 mo)[6] | $RR_{Death\_IfBRD}=1.78$ (Up to 6 weeks age)/1.09[7] | .. | [33] |

[1] : Diseases labels respectively stands for: Dys: Dystocia, FMR: Foetal membranes retention, Met: Metritis, PVD: Endometritis, FIT: Failure of passive immune transfer, Dia: Diarrhoea, Omph: Omphalitis, Sept: Septicaemia, BRD: respiratory diseases.

[2] : Risk is the basal value of risk out of any risk factor, 5% weekly standard deviation was applied. For cows diseases weeks of occurrence stands for milking week and for ages for calves diseases.

[3] : ADG decrease effect of a disease per day and per animal (duration of effect in weeks).

[4] : Relative risk on conception rate decrease (duration of effect in weeks).

[5] : 6 weeks effect duration if first infection, 2 weeks if second case or more. Effects is applied during 24 weeks starting for the week of infection).

[6] : Relative risk for fist case/second/ third or more.

[7] : Relative risk applied to death risk applied until 1-year age.

Preventive treatments (vaccines) were considered for diarrhoea and respiratory diseases, with two vaccination strategies (Table 5) that has been associated with decrease in incidence (ImpactInci$_{BRD}$; ImpactInc$_{Dia}$) and changes in the consequences of diarrhoea and BRD (e.g., ImpactDeath$_{BRD}$ and ImpactDeath$_{Dia}$ for mortality reduction through vaccination against respiratory diseases and diarrhoea, respectively). However, vaccination did not reduce treatment duration compared to non-vaccinated calves. In the case of FIT, the diarrhoea vaccine was considered as ineffective (ImpactInc$_{Dia}$ = 1 and ImpactMort$_{Dia}$ = 1).

To simulate pre-vaccination prevalences, diarrhoea and BRD risks (Risk$_{BRD}$ and Risk$_{Dia}$) were adjusted by coefficients of 0.5 (low infection risk) and 1.5 (high infection risk).

Antibiotic use was evaluated as indicated in [Eq. 1] through the Animal Level of Exposure to Antimicrobials (ALEA), which is a reference indicator in France for monitoring animal exposure to antibiotics.

$$ALEA \ = \ Treated\ bodyweight\ with\ AM \ / \ Treatable\ bodyweight \quad\quad [Eq.\ 1]$$

## Economic assessment of vaccination

The economic evaluation included revenues provided by farm products (Total_Revenue), calculated by considering the revenues from the sale of fattened male calves (MaleSold), fattened heifers (FemaleSold), and culled cows intended for herd renewal (CullSold), as indicated in [Eq. 2].

$$Total\_Revenue\ (€) \ = \ Revenue_{FemaleSold} \ + \ Revenue_{MaleSold} + \ Revenue_{CullSold} \quad\quad [Eq.\ 2]$$

Variable costs encompassed feed costs (FeedCost$_{Total}$), medications (Cost$_{Med}$), veterinary interventions (Cost$_{Vet}$), and vaccines (Cost$_{Vac}$), as indicated in [Eq. 3].

$$Total\_Cost\ (€) \ = \ FeedCost_{Total} + \ Cost_{Med} + \ Cost_{Vet} + \ Cost_{Vac} \quad\quad [Eq.\ 3]$$

The annual Net Value of Farm Production (NV$_{Prod}$) was calculated as indicated in [Eq. 4], excluding expenditures that remains the same across years (such as costs across scenarios) and calculated techno-economic indicators such as the feed cost per fattened male calf (FeedCost$_{Male}$) or per fattened heifer (FeedCost$_{Female}$).

**Table 5. Vaccination program and vaccine characteristics (SPC[1] and experts' data).**

|  | Vaccine protocol (animal, route[2] and period, cost per dose and time for administration) | Impact on disease incidence[*] | Impact on disease severity[*] |
|---|---|---|---|
| **Dia** | Rotavec® (cows; IM; 3–12 wks before calving; €10; 5 min) | ImpactInci$_{Dia}$ = 0.7 (1 if FIT[3]) | ImpactDeath$_{Dia}$ = 0.7 ImpactADG$_{Dia}$ = 0.7 |
| **BRD** | Rispoval® (calf; IN; 1 week old; €6; 3 min) Bovigrip® (calf; IM; 8 wks old; €6; 4 min) Bovigrip® (calf; IM; 12 wks old; €6; 4 min) | ImpactInci$_{BRD}$ = 0.5 | ImpactDeath$_{BRD}$ = 0.5 ImpactADG$_{BRD}$ = 0.5 |

[1] : Summary of Product Characteristics (SPC) for authorized veterinary medicines [accessible at https://www.ircp.anmv.anses.fr/ consulted on 17/01/2025].

[2] : IM: intramuscular; IN: intranasal.

[3] : Failure Immune Transfer.

[*] : The impact on incidence is applied as a reduction effect of the simulated incidence risk for simulated animals (Dia: for the incidence of diarrhea and BRD for the incidence of respiratory diseases). The effect of the Dia vaccine is cancelled in case of failure of immunity transfer (Impact = 1). The impact on severity is a reduction effect of the consequences of production losses (Table 4).

$$NV_{Prod} (€) = Total\_Revenue - Total\_Cost \qquad \text{[Eq. 4]}$$

The net value of vaccination ($NV_{Vacc}$) was the difference in $NV_{Prod}$ when vaccination occurred compared to no vaccination, as indicated in [Eq. 5].

$$NV_{Vacc} (€) = NV_{Prod\_Vaccination} - NV_{Prod\_NoVaccination} \qquad \text{[Eq. 5]}$$

The return on investment (ROI) of vaccination was then calculated as the net value per euro spent on vaccine, as indicated in [Eq. 6]

$$ROI = NV_{Vacc} / Cost_{Vac} \qquad \text{[Eq. 6]}$$

The labour of the farmer was considered for care activities, animal handling and vaccination. The marginal gain from an hour worked on care tasks, assistance with care, and vaccination of the studied diseases was then calculated as indicated in [Eq. 7], to estimate the value of labour for the farmer while adopting a vaccination scenario.

$$Labor\_marginal\_gain (€/hour) = (Labor_{Vaccination} - Labor_{NoVaccination}) / (NV_{Prod\_Vaccination} - NV_{Prod\_NoVaccination}) \qquad \text{[Eq. 7]}$$

## Simulation and calibration

The simulation model was run for 36 scenarios, combining:

- Four vaccination levels (none, BRD-only, diarrhoea-only, both vaccines combined),

- Three levels of risk factors associated with the deterioration or improvement of the baseline situation, for each of the two diseases (totaling 9 levels).

Each scenario simulation lasted 520 weeks and was repeated 100 times. The average results per year were analyzed. The results were presented in the form of indicators that allowed both model validation and analysis of the parameters of interest (S3 Table in S1 File).

The calibration (epidemiological and economic parameters) was detailed in Tables 1–5. The treatments used and the probabilities of intervention were reported in S2 Table in S1 File, and medication characteristics were provided in S3 Table in S1 File.

To account for financial risk, all product sale prices and feed costs followed a normal distribution of ±5% each year for each simulation.

## Results

### Model validation for herd dynamics reproduction

The modeled herd consisted of an average of 99.67 (±1.61) cows (HerdSize) with a replacement rate (replacementRate) of 26.03% (±1.35%) across all scenarios (S1 Fig in S1 File). The first calving (AgeCalv1) occurred at 161 weeks (3 years and 6 weeks), and the calving interval (CalvInter) was 396.84 days (±0.98 days). The culling rate (CullingRate) and cow mortality rate (MorRate) were 24.²26% (±2.06%) and 2.01% (±0.14%), respectively.

On average, 55.18 (±0.92) male calves ($Nbirth_{Male}$) and 55.21 (±0.91) female calves ($Nbirth_{Female}$) were born annually, yielding a total of 110.39 (±0.13) annual calvings. The number of vaccines administered each year averaged 110.47 (±0.14) for diarrhoea (±1 dose per animal) and 311.32 (±1.27) for BRD (3 doses per animal).

In line with the calibration assumptions regarding farmer behaviour, the selling weights with or without vaccination were similar: 385.77 kg (±0.51 kg) for males and 308.97 kg (±1.55 kg) for females, respectively. The difference between the target weights in the calibration (Table 1) and the modeled weights (S1 Table in S1 File) resulted from the delay between the decision-making point (reaching the target weight) and the actual sale weight, which accounted for average daily weight gain during the interim period.

The incidences of dystocia (Dys), foetal membrane retention (FMD), metritis (Met) and endometritis (PVD) were 11.33% (±0.11%), 4.21% (±0.07%), 2.98% (±0.04%), and 5.94% (±0.08%), respectively, with no differences observed between vaccination groups. In calves, the prevalence of omphalitis (Omph) and septicaemia (Sept) were 17.61% (±0.13%) and 2.73% (±0.06%). respectively. These results were consistent across all vaccination scenarios.

### Epidemiological outcomes and antibiotic use per vaccination scenario

Tables 6 and 7 present the key results across vaccination scenarios, considering average baseline risks (Table 4) and risks adjusted by coefficients of 0.5 (low infection risk) and 1.5 (high infection risk). The prevalence of diarrhoea and BRD (Table 6: Prev$_{Dia}$, Prev$_{BRD}$) ranged from 8%–33% and 13%–79%, respectively, depending on vaccination groups and disease incidence levels. On average, vaccination against diarrhoea and respiratory diseases reduced the prevalences by 24% and 50%, respectively (S2 and S4 Figs in S1 File).

The weights of males and females at selling (S1 Table in S1 File: MalWeight, FemWeight) varied slightly across scenarios due to selling at fixed target weights. However, the selling ages (Table 6: AgeMal, AgeFem) decreases by 0.3% to 3% with vaccination scenarios.

Vaccination was associated with a consistent reduction in animal exposure to antibiotics, whatever the scenario, with more significant ALEA reductions observed in BRD vaccination scenarios due to the larger decrease in BRD prevalence and the higher systematic reliance on antibiotics for respiratory disease treatment (as per calibration assumptions, Table 1).

**Table 6. Output of the bioeconomic models: Prevalence, the age of sale of animals, and exposure to antibiotics.**

| Scenarios[1] | | AgeMal[3] (wk) | | AgeFem[3] (wk) | | Prev$_{Dia}$ | | Prev$_{BRD}$ | | ALEA_calves$_{1m}$ | | ALEA_calves$_{6m}$ | | ALEA_allcalves | |
|---|---|---|---|---|---|---|---|---|---|---|---|---|---|---|---|
| Risk | Vacc | Mean[2] | Δ%Ref[2] | Mean[2] | Δ%Ref[2] | Mean[2] | Δ%Ref[2] | Mean[2] | Δ%Ref[2] | Mean[2] | Δ%Ref[2] | Mean[2] | Δ%Ref[2] | Mean[2] | Δ%Ref[2] |
| **Basic** | 00 | 43.35 | ·· | 42.57 | ·· | 0.22 | ·· | 0.52 | ·· | 2.22 | ·· | 1.59 | ·· | 0.24 | ·· |
| | 0R | 42.70 | −1.50% | 41.60 | −2.29% | 0.22 | ·· | 0.26 | −50.9% | 1.68 | −24.19% | 1.19 | −24.93% | 0.20 | −17.79% |
| | D0 | 43.20 | −0.35% | 42.31 | −0.62% | 0.17 | −23.7% | 0.53 | ·· | 2.13 | −4.15% | 1.52 | −4.16% | 0.23 | −5.41% |
| | DR | 42.57 | −1.80% | 41.44 | −2.66% | 0.17 | −22.5% | 0.26 | −49.8% | 1.61 | −27.70% | 1.14 | −28.53% | 0.19 | −22.20% |
| **0.5** | 00 | 42.89 | ·· | 41.95 | ·· | 0.11 | ·· | 0.26 | ·· | 1.47 | ·· | 1.04 | ·· | 0.17 | ·· |
| | 0R | 42.64 | −0.58% | 41.56 | −0.94% | 0.11 | ·· | 0.13 | −51.0% | 1.22 | −17.56% | 0.85 | −18.55% | 0.15 | −12.06% |
| | D0 | 42.83 | −0.14% | 41.85 | −0.26% | 0.08 | −24.1% | 0.26 | ·· | 1.41 | −4.09% | 1.00 | −3.94% | 0.16 | −4.88% |
| | DR | 42.56 | −0.77% | 41.48 | −1.14% | 0.08 | −24.8% | 0.13 | −50.3% | 1.16 | −21.18% | 0.81 | −22.05% | 0.14 | −16.25% |
| **1.5** | 00 | 43.91 | ·· | 43.17 | ·· | 0.33 | ·· | 0.79 | ·· | 2.93 | ·· | 2.13 | ·· | 0.31 | ·· |
| | 0R | 42.78 | −2.56% | 41.73 | −3.34% | 0.33 | ·· | 0.39 | −50.9% | 2.17 | −26.02% | 1.55 | −27.31% | 0.25 | −19.16% |
| | D0 | 43.69 | −0.49% | 42.78 | −0.91% | 0.25 | −23.1% | 0.79 | ·· | 2.80 | −4.70% | 2.02 | −5.24% | 0.29 | −6.12% |
| | DR | 42.56 | −3.07% | 41.42 | −4.06% | 0.25 | −23.1% | 0.39 | −50.7% | 2.03 | −30.75% | 1.45 | −32.07% | 0.23 | −25.36% |

[1] The column of scenarios represents the infection risk scenarios and the vaccination scenarios. Basic: base risks from the calibration; 0.5: a relative risk of 0.5 was applied to the risks of Dia and BRD; 1.5: a relative risk of 1.5 was applied to the risks of Dia and BRD; 00: scenario where no vaccination protocol was implemented; DR: scenario of implementing the vaccination protocol for Dia (D) and BRD (R).

[2] The average result of the indicator per scenario is presented in the Mean columns and the variation of this result compared to the reference scenario is presented in the Δ%Ref columns. The reference scenario is the scenario without vaccination (00)

[3] AgeMal stands for the simulated age of sale of male calves, AgeFem stands for the simulated age of sale of femal calves.

Mortality was consistently higher in males than females (S4 Table in S1 File). Vaccination reduced mortality during the 2-day to 1-month (2d-1mo) period by 2–3% for diarrhoea and 15–17% for BRD. This reduction was more pronounced in younger calves than in older ones. Additionally, the reduction in mortality varied non-linearly with baseline risk levels and the sex of the animals:

- In males, the mortality reduction due to vaccination was stronger for medium or high baseline risks (coef 1.5) than for low baseline risks (coef 0.5);

- In females, the mortality reduction due to vaccination was similar for medium or low baseline risks (coef 0.5) but significantly higher for high baseline risks (coef 1.5).

### Economic results by vaccination scenario

Vaccination against respiratory diseases reduced average feed cost of sold animals (FeedCost$_{Male}$, FeedCost$_{Female}$) by 1% to 5%, with greater reductions observed for females. This was attributed to shorter fattening durations, ranging from 0.4 to 1.44 weeks (S5 Table in S1 File). Vaccination against diarrhoea had a smaller impact on feed costs and fattening durations (−0.28% to −1.34%). All vaccination scenarios also led to reductions in disease control costs (Cost$_{Vet}$, Cost$_{Med}$) of 4% to 22% (€43–€272/year) (S5 Table in S1 File).

The overall net value of vaccination was positive (Table 7), except for scenarios with low infection risks for diarrhoea and BRD (Coeff. of 0.5). The return on investment (ROI) for vaccination was approximately 0.4–0.5 (€1 invested returned €0.4–0.5) in the baseline scenario (medium risk), negative for low infection risks, and approaches 1 (€1 invested returned €1) for high infection risks (Coeff. 1.5). For medium infection risks, the labour marginal gain was €58–€75, reflecting the earnings per extra hour of work dedicated to vaccination. For high infection risks, this value increased to €150–€220 per hour. Vaccination also reduced antimicrobial use and increased farm revenue (Table 7: NV$_{Vacc}$) with an estimated €1,329–€5,835 of additional revenue per unit reduction in ALEA. Economic outcomes were highly sensitive to calibration parameters.

**Table 7. Output of the bioeconomic models: Economic results.**

| Scenarios[1] | | Labor (h/yr) | | Total Revenue (€) | | Total_Cost (€) | | NV$_{Prod}$ (€) | | Cost$_{Vac}$ | NV$_{Vacc}$ (€) | ROI | €/h | €/Δ$_{ALEA}$ | h/Δ$_{ALEA}$ |
|---|---|---|---|---|---|---|---|---|---|---|---|---|---|---|---|
| Risk | Vacc | Mean[2] | Δ%Ref[2] | Mean[2] | Δ%Ref[2] | Mean[2] | Δ%Ref[2] | Mean[2] | Δ%Ref[2] | | | | | | |
| Basic | 00 | 32.5 | ·· | 139854 | ·· | 48672 | ·· | 91181 | ·· | 0 | ·· | ·· | ·· | ·· | ·· |
| | 0R | 44.8 | 38% | 141397 | 1.10% | 49501 | 1.70% | 91895 | 0.78% | 1862 | 714 | 0.38 | 58.08 | −1329 | −22.9 |
| | D0 | 39.8 | 22% | 141378 | 1.09% | 49658 | 2.03% | 91719 | 0.59% | 1099 | 538 | 0.49 | 74.17 | −5835 | −78.7 |
| | DR | 52.2 | 61% | 142980 | 2.24% | 50503 | 3.76% | 92477 | 1.42% | 2974 | 1295 | 0.44 | 65.66 | −2105 | −32.1 |
| 0.5 | 00 | 21.7 | ·· | 142445 | ·· | 47507 | ·· | 94937 | ·· | 0 | ·· | ·· | ·· | ·· | ·· |
| | 0R | 37.4 | 72% | 142211 | −0.16% | 48723 | 2.56% | 93488 | −1.53% | 1868 | −1449 | −0.78 | −92.44 | 5597 | −60.6 |
| | D0 | 29.8 | 37% | 143432 | 0.69% | 48584 | 2.27% | 94848 | −0.09% | 1104 | −89 | −0.08 | −11.11 | 1488 | −133.9 |
| | DR | 45.5 | 109% | 143776 | 0.93% | 49861 | 4.95% | 93915 | −1.08% | 2981 | −1022 | −0.34 | −42.99 | 3274 | −76.2 |
| 1.5 | 00 | 43.4 | ·· | 138384 | ·· | 50021 | ·· | 88363 | ·· | 0 | ·· | ·· | ·· | ·· | ·· |
| | 0R | 52.5 | 21% | 140580 | 1.59% | 50292 | 0,54% | 90287 | 2.18% | 1856 | 1924 | 1.04 | 211.34 | −2522 | −11.9 |
| | D0 | 49.6 | 14% | 140190 | 1.31% | 50906 | 1,77% | 89284 | 1.04% | 1100 | 921 | 0.84 | 150.43 | −6677 | −44.4 |
| | DR | 59.0 | 36% | 143043 | 3.37% | 51258 | 2,47% | 91784 | 3.87% | 2979 | 3421 | 1.15 | 219.69 | −3793 | −17.3 |

[1] The column of scenarios represents the infection risk scenarios and the vaccination scenarios. Basic: base risks from the calibration; 0.5: a relative risk of 0.5 was applied to the risks of Dia and BRD; 1.5: a relative risk of 1.5 was applied to the risks of Dia and BRD; 00: scenario where no vaccination protocol was implemented; DR: scenario of implementing the vaccination protocol for Dia (D) and BRD (R).

[2] The average result of the indicator per scenario is presented in the Mean columns and the variation of this result compared to the reference scenario is presented in the Δ%Ref columns. The reference scenario is the scenario without vaccination (00).

 

## Discussion

### Validation of the method and added value of the bioeconomic model

National control strategies for animal production diseases yield inconsistent economic outcomes depending on the production system and the estimated prevention and control measures. Thus, farm-level modelling remains the most suitable approach for assessing the economic impacts of these diseases and their management strategies [34–36]. Several farm-level models have been developed to represent cow-calf systems, employing various techniques such as partial stochastic mechanistic modelling or linear programming [22,37]. The bio-economic stochastic model developed in this study adopts a dynamic, mechanistic, and holistic framework to represent a French cow-calf farming system with Charolais cattle. For multifactorial diseases, and given the common challenges associated with adopting preventive practices, it is essential to conduct an economic evaluation that extends beyond purely monetary aspects [27].

The bio-economic model used here integrates both epidemiological and economic components, enabling a comprehensive analysis of production and health outcomes in a French cow-calf herd. The results demonstrate the model's accuracy in reproducing herd dynamics, as evidenced by key output indicators, including herd size, epidemiological trends, production metrics, and technical parameters. For medium-risk scenarios, the model outputs were validated against official performance data for French cow-calf systems [38,39]. The ALEA values generated by the model align with established literature (Table 6). For calves under one month of age (ALEA_calves$_{1m}$), a value of approximately 2.2 is obtained, consistent with reports for veal production systems. Similarly, the ALEA for the entire French cattle population (ALEA_allcalves) closely corresponds to empirical data when all calves and replacement heifers are considered. As expected, ALEA values decrease with age (ALEA_calves$_{1m}$, ALEA_calves$_{6m}$ and ALEA_allcalves) due to a decline in the number of treatments (numerator) and an increase in total live weight (denominator).

The model also accurately reproduces calf mortality rates across all scenarios, yielding results consistent with national data (mortality at 0–6 months at 8–10%) (S4 Table in S1 File; S3 Fig in S1 File). Despite the relative simplicity of beef production systems, predictive accuracy remains challenging, similar to the complexities encountered in dairy farming. Biological variability, economic uncertainty, and calibration accuracy represent significant challenges. Expanding the number of scenarios to capture diverse farm conditions is particularly relevant.

The economic evaluation does not consider the loss in carcass monetary value of an animal that has been infected, associated with conformation issues relative to slaughterhouse standards. The study also does not consider broader economic effects, such as potential market impacts on exports due to a comparative advantage held by disease-free farms. These aspects fall outside the scope of the current study [40].

### Economic assessment of vaccination

The model indicates a positive net value of vaccination (excluding labour costs) for the majority of farms, with the exception of those operating under low infectious risk conditions for diarrhoea and BRD. Since labour costs are excluded from the financial analysis, it is critical to evaluate the economic impact of vaccination in conjunction with the associated additional labour. This consideration is particularly relevant because the total labour time required for vaccination and treatment activities increases significantly under vaccination protocols (Table 7: Labour). The theoretical valuation of labour (expressed as additional euros earned per extra hour of work) facilitates the simultaneous consideration of these two dimensions. The results reveal that most farms achieve favourable returns on the additional labour associated with vaccination strategies. For instance, under medium infection risk scenarios, farmers earn €58–75 more per additional hour of labour allocated to vaccination, with this figure rising to €150–220 under high infectious risk conditions.

The monetary valuation of ALEA reductions (Table 7: €/Δ$_{ALEA}$) aligns with findings in the existing literature, which report substantial variability in the marginal cost of ALEA reduction depending on the ALEA value and farm-specific characteristics [41,42]. The results presented in Table 7 show that the marginal cost of ALEA reduction (the inverse of the €/Δ$_{ALEA}$

indicator) does not account for labour costs but clearly illustrates cases where this cost is negative, signifying concurrent monetary gains and ALEA reductions. This variability in ALEA reduction outcomes across different risk levels is well-documented in the literature [42,43]. Although the economic results are highly sensitive to scenario and calibration parameters, they consistently demonstrate the link between vaccination and economic benefits for farmers in most cases.

The economic results also underscore the advantages of vaccination as a profitable investment, particularly for farms with moderate to high baseline disease risks. However, the return on investment remains relatively modest, totalling €0.30 per euro invested (excluding labour costs. For farms with low infectious risk levels for diarrhoea and BRD, vaccination does not appear economically viable under the current calibration assumptions but may act as a form of insurance against potential health risks. These findings are consistent with prior studies. For example, the ROI for diarrhoea vaccination was found to range from €0.49 to €0.84, depending on the incidence risk. Arnoux et al. (2021) estimated the annual expected reward of BVD vaccination at €0.80 under high-risk conditions in French Charolais cow-calf systems [25].

The heterogeneity in vaccination rewards across and within scenarios has been identified in the literature as a key factor explaining the divergence in farmers' attitudes towards vaccination [25]. For BRD, there is a notable disparity between the measurable effects of vaccination and its perception among French livestock farmers. Despite documented benefits, such as an improved average daily carcass gain in animals vaccinated against BRSV, many farmers perceive vaccination as an additional cost and a logistical challenge in pasture-based cow-calf systems [44,45]. This perception is likely influenced by the availability of alternative strategies for managing BRD, including preconditioning animals [46], enhancing biosecurity measures [47], implementing quarantine protocols, and isolating sick animals from the main herd [48]. These practices complicate the identification of direct benefits that are specific to vaccination. We have emphasized that the benefits of vaccination are strongly influenced by the epidemiological context, while also showing significant variation within a specific context. This variability in benefits could potentially explain why farmers' approaches to major diseases control are so diverse in practice [25]. To better capture farmers' strategic behaviour and evaluate large-scale collective control programs, it is essential to develop models that integrate farmers' decision-making processes within the dynamics of the epidemiological situation [49]. Overall, despite the significant benefits associated with vaccination, including reductions in medication, veterinary intervention, and feed costs, the cost of vaccines and the lack of financial compensation does not give farmers any economic incentive to implement systematic vaccination practices [45]. (S5 Table in S1 File).

Our findings suggest that the primary argument for vaccination lies in increased production efficiency rather than in cost reductions alone. To effectively encourage farmers to adopt vaccination practices, it is essential to emphasise not only the potential cost savings but also the long-term advantages that vaccination offers. These benefits include improved long-term herd health, enhanced productivity, and a stronger reputation, which may lead to better market access. Additionally, the cost-effectiveness of vaccination should be highlighted through a multicriteria evaluation approach, integrating farm-level economic outcomes, the role of vaccination in reducing antimicrobial resistance, and the financial value of additional labour, particularly in family farming systems. Such an approach would underscore the economic and societal rationale for vaccination, benefiting both livestock farmers and public health.

## Supporting information

**S1 File. S1 Table: List of calculated indicators.** S2 Table: Treatment protocol for cows and calves. S3 Table: Characteristics of medicines used for cows and calves' diseases. S4 Table: Other output results of the bioeconomic model: mortality rates. S5 Table: Other output results of the bioeconomic model: other economic outputs of the model. S1 Fig: Other output results of the bioeconomic model: average animal presence per year and per scenario. S2 Fig: Other output results of the bioeconomic model: Graphical representation of the reduction in the prevalence of respiratory diseases and diarrhea. S3 Fig: Other output results of the bioeconomic model: Graphical representation of the reduction in calf's mortality

rates. S4 Fig: Average simulated annual cases of Diarrhea and Respiratory Diseases across scenarios with and without vaccination.
(DOCX)

## Author contributions

**Conceptualization:** Ahmed Ferchiou, Didier Raboisson.

**Data curation:** Ahmed Ferchiou, Nicolas Herman, Didier Raboisson.

**Formal analysis:** Ahmed Ferchiou, Anthony Giacomini, Nicolas Herman.

**Funding acquisition:** Guillaume Lhermie, Didier Raboisson.

**Investigation:** Ahmed Ferchiou.

**Methodology:** Ahmed Ferchiou, Didier Raboisson.

**Project administration:** Guillaume Lhermie, Didier Raboisson.

**Resources:** Guillaume Lhermie, Didier Raboisson.

**Software:** Ahmed Ferchiou.

**Supervision:** Ahmed Ferchiou, Didier Raboisson.

**Validation:** Nicolas Herman, Guillaume Lhermie, Didier Raboisson.

**Visualization:** Ahmed Ferchiou, Anthony Giacomini, Didier Raboisson.

**Writing – original draft:** Ahmed Ferchiou, Anthony Giacomini, Didier Raboisson.

**Writing – review & editing:** Ahmed Ferchiou, Anthony Giacomini, Guillaume Lhermie, Didier Raboisson.

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
