## [Decision Letter · Decision Letter 0]

21 Mar 2025

Dear Dr. Ferchiou,

Both reviewers found merit in your manuscript, and I concur. However, they have highlighted areas where changes are needed. Please read the reviewers' comments below and in the attached file and make revisions to the manuscript accordingly to address these concerns.

We look forward to receiving your revised manuscript.

Kind regards,

Angel Abuelo, DVM, MRes, MSc, PhD, DABVP (Dairy), DECBHM

Academic Editor

PLOS ONE

Journal Requirements:

2. We note that there is identifying data in the Supporting Information file <BHS_diarrhea_bovine_resp_diseases-Sup_Mat_v2025-01-20.docx>. Due to the inclusion of these potentially identifying data, we have removed this file from your file inventory. Prior to sharing human research participant data, authors should consult with an ethics committee to ensure data are shared in accordance with participant consent and all applicable local laws.

-Location data

Please remove or anonymize all personal information (Name), ensure that the data shared are in accordance with participant consent, and re-upload a fully anonymized data set. Please note that spreadsheet columns with personal information must be removed and not hidden as all hidden columns will appear in the published file.

Reviewers' comments:

Reviewer's Responses to Questions

**Comments to the Author**

1. Is the manuscript technically sound, and do the data support the conclusions?

Reviewer #1: Yes

Reviewer #2: Yes

2. Has the statistical analysis been performed appropriately and rigorously?

Reviewer #1: N/A

Reviewer #2: Yes

3. Have the authors made all data underlying the findings in their manuscript fully available?

Reviewer #1: Yes

Reviewer #2: Yes

4. Is the manuscript presented in an intelligible fashion and written in standard English?

Reviewer #1: Yes

Reviewer #2: Yes

Reviewer #1: This manuscript attempts to measure the economics of vaccinating beef animals in France. The basic simulation model is fairly standard and appears to be done correctly. However, the authors oversell the economic benefits of vaccinating calves and cows by focusing on 1 economic measure, return on investment (ROI). Using this measure alone appears to make the vaccinating of these animals as "feasible" using ROI, but ROI is not the only measure farmers will use to make decisions. When broken down the different vaccination scenarios have marginal impacts on revenue and or cost, ranging from <1% to < 4% for revenue and from 2.5% to 4% increases in costs, leading to between 1 and 4% increases in net revenue. Thus, one must ask do these numbers make the proposed vaccination "really" profitable when one factors in time and labour for farmers to undertake the vaccination programs presented, particularly in the low risk scenarios.

A minor comment/question is Prod_revenue in equation 4, the same as Total_Revenue in equation 2? If not explain why? If so then it needs to be changed in either equation for consistnecy.

Reviewer #2: This is an interesting and insightful paper. While the paper is well-written. and comprehensive, there are key areas that the authors ought to consider to improve the robustness of their analysis and the theoretical contributions of the paper.

First, authors should highlight what differentiate the paper (esp. the model) from other research that looked at the same topic. By doing so, the research gap will be explicitly highlighted to show the theoretical contribution of the paper. Second, authors may have to consider differentiating the herd based on the health status - susceptible, infected, recovered, and vaccinated. Such segmentation will allow authors to highlight the morbidity (carcass value) losses, which is currently not captured in. the model.

Lastly, graphical representation of the results will provide more insights into the disease dynamics.

**Do you want your identity to be public for this peer review?** For information about this choice, including consent withdrawal, please see our Privacy Policy

Reviewer #1: No

Reviewer #2: **Yes: ** Joshua Aboah

---

## [Author Response · Author response to Decision Letter 1]

17 Apr 2025

All comments have been addressed. Graphs have been added as recommended, as well as the positioning of the model in relation to the existing literature. Regarding the integration of morbidity-related losses, the section that caused confusion has been rephrased, and we clarified in our response how morbidity losses are incorporated into the model.

We thank the reviewers for their efforts and the suggestions that have improved the quality of our manuscript.

---

## [Decision Letter · Decision Letter 1]

28 May 2025

Dear Dr. Ferchiou,

Thank you for submitting your manuscript to PLOS ONE. After careful consideration, we feel that it has merit but does not fully meet PLOS ONE’s publication criteria as it currently stands. Therefore, we invite you to submit a revised version of the manuscript that addresses the points raised during the review process.

We look forward to receiving your revised manuscript.

Kind regards,

Angel Abuelo, DVM, MRes, MSc, PhD, DABVP (Dairy), DECBHM

Academic Editor

PLOS ONE

Journal Requirements:

Reviewers' comments:

Reviewer's Responses to Questions

**Comments to the Author**

Reviewer #1: (No Response)

Reviewer #2: All comments have been addressed

2. Is the manuscript technically sound, and do the data support the conclusions?

Reviewer #1: Yes

Reviewer #2: Yes

3. Has the statistical analysis been performed appropriately and rigorously?

Reviewer #1: Yes

Reviewer #2: Yes

4. Have the authors made all data underlying the findings in their manuscript fully available?

Reviewer #1: Yes

Reviewer #2: No

5. Is the manuscript presented in an intelligible fashion and written in standard English?

Reviewer #1: No

Reviewer #2: Yes

Reviewer #1: The revisions to the manuscript are acceptable, however the format of tables needs a lot of work. Tables are required to stand alone without the reader having to refer to the text to understand the content therein. The authors use variable names that are incoherent without referring to the text, and many columns have no heading, thus the reader is left to try and work out what the numbers in the table are, whether they are monetary values or percentages. These need to be corrected before the manuscript is suitable for publication

Reviewer #2: The authors have adequately addressed the previously raised concerns. However, the manuscript lacks figures to confirm whether the graphical trends of certain results align with the reported findings. I recommend including appropriate figures to illustrate these trends for clarity and completeness.

Aside from this, I am satisfied with the current version of the manuscript.

**Do you want your identity to be public for this peer review?** For information about this choice, including consent withdrawal, please see our Privacy Policy

Reviewer #1: No

Reviewer #2: No

---

## [Author Response · Author response to Decision Letter 2]

11 Jul 2025

Reviewer #1:

The revisions to the manuscript are acceptable, however the format of tables needs a lot of work. Tables are required to stand alone without the reader having to refer to the text to understand the content therein. The authors use variable names that are incoherent without referring to the text, and many columns have no heading, thus the reader is left to try and work out what the numbers in the table are, whether they are monetary values or percentages. These need to be corrected before the manuscript is suitable for publication

Authors :

Thank you for this comment. We have revised the two calibration tables for diseases and vaccination scenarios, as well as the two result tables. We believe that Tables 1 to 3 are now clear and comprehensive.

Table 4: Some labels have been reformulated and explanatory footnotes have been added or clarified.

Table 5: An explanatory footnote on vaccine effects has been included.

Tables 6 and 7: The missing header row has been added. For several indicators, both the simulation result (unit of the indicator) and its variation relative to the reference scenario are now provided. This clarification should address the confusion between absolute and percentage values.

Reviewer #2:

The authors have adequately addressed the previously raised concerns. However, the manuscript lacks figures to confirm whether the graphical trends of certain results align with the reported findings. I recommend including appropriate figures to illustrate these trends for clarity and completeness.

Aside from this, I am satisfied with the current version of the manuscript.

Authors :

Thank you for your comment.

For this manuscript, we opted for a primarily tabular representation to facilitate the simultaneous reading of various zootechnical, epidemiological, and economic indicators.

We agree that the paper lacks graphical representations. A figure showing the average herd size was included for the validation of simulated herd dynamics (S1 Figure 1). In response to your comment, we have now added figures illustrating the simulated annual number of diarrhea and respiratory disease cases, with and without vaccination (Dia and BRD) respectively (S1 Figure 4).

If your comment refers to the trends for each individual simulation within a given scenario, unfortunately we are unable to display them without re-running the model (which would require adjustments to the code), as we only export the annual means and standard deviations for each scenario (shown in S1 Table 1 for the general indicators).

---

## [Editor Report · Decision Letter 2]

15 Jul 2025

Economics of vaccination against diarrhoea and respiratory diseases in French cow calves’ systems: a modelling approach

PONE-D-25-05016R2

Dear Dr. Ferchiou,

We’re pleased to inform you that your manuscript has been judged scientifically suitable for publication and will be formally accepted for publication once it meets all outstanding technical requirements.

Kind regards,

Angel Abuelo, DVM, MRes, MSc, PhD, DABVP (Dairy), DECBHM

Academic Editor

PLOS ONE
---

## [Editor Report · Acceptance letter]

PONE-D-25-05016R2

PLOS ONE

Dear Dr. Ferchiou,

I'm pleased to inform you that your manuscript has been deemed suitable for publication in PLOS ONE. Congratulations! Your manuscript is now being handed over to our production team.

Kind regards,

on behalf of

Dr. Angel Abuelo

Academic Editor

PLOS ONE